# Challenges in Multiple Myeloma Therapy in Older and Frail Patients

**DOI:** 10.3390/cancers17060944

**Published:** 2025-03-11

**Authors:** Anna Aureli, Beatrice Marziani, Tommaso Sconocchia, Gianmario Pasqualone, Luca Franceschini, Giulio Cesare Spagnoli, Adriano Venditti, Giuseppe Sconocchia

**Affiliations:** 1CNR Institute of Translational Pharmacology, 67100 L’Aquila, Italy; giuliocesare.spagnoli@ift.cnr.it; 2Emergency Medicine Department, Sant’Anna University Hospital, Via A. Moro, 8, Cona, 44124 Ferrara, Italy; mrzbrc@unife.it; 3San Raffaele Telethon Institute for Gene Therapy (SR-TIGET), IRCCS San Raffaele Scientific Institute, Via Olgettina 58, 20132 Milano, Italy; sconocchia.tommaso@hsr.it; 4Department of Biomedicine and Prevention, University of Rome Tor Vergata, 00133 Rome, Italy; gianmario.pasqualone@uniroma2.it (G.P.); luca.franceschini@uniroma2.it (L.F.); adriano.venditti@uniroma2.it (A.V.)

**Keywords:** multiple myeloma, frailty, immunotherapy, CAR-T, bispecific antibodies

## Abstract

MM incidence is increasing globally, particularly in high-income countries and in the male population aged ≥ 50. While advances in treatment strategies have led to improvement of survival over the past decades, MM remains incurable in a large percentage of cases. Importantly, about one third of patients diagnosed with MM are >75 years old and are characterized by relatively low fitness or frailty. Few older and frail patients diagnosed with multiple myeloma (MM) are currently enrolled in pivotal clinical trials, and little data are presently available on appropriate management. These patients are more vulnerable to stressors and present lower resistance to cancer and related treatments. Here, we reviewed relevant studies analyzing myeloma treatments in frail and older patients, and we analyzed their effectiveness and safety.

## 1. Introduction

Multiple myeloma (MM) is an aggressive hematologic malignancy characterized by an excess of plasma cells (PCs) in the bone marrow (BM). Its progression starts with a pre-malignant monoclonal gammopathy of undetermined significance (MGUS) [1], followed by an intermediate asymptomatic stage, smoldering multiple myeloma (SMM) [2], which preludes MM development in high percentages of patients [2].

Malignant transformation begins inside lymph nodes’ germinal centers, where B-lymphocytes undergo genetic and epigenetic alterations before moving into BM, where they differentiate into PCs by interacting with the immune microenvironment [3]. Malignant PCs produce serum immunoglobulins and activate osteoclasts, resulting in bone tissue degradation, and focal injuries, leading to bone fractures [4].

Cytogenetic anomalies such as immunoglobulin heavy-chain locus translocations and hyperdiploid karyotypes are already detectable in premalignant phases [5], but only 1% of MGUS cases progress to MM [6], suggesting that the switch to symptomatic disease may be due to sudden, low-frequency, randomly acquired genomic events [7].

While progression to SMM does not usually produce end-organ damage [8], in some cases, MM features might already be detectable [9] with malignant plasma cell levels > 10% in the BM and unusually large numbers of “M spike” serum proteins. The weakening of the immune system, anemia, bleeding disorders, hypercalcemia, kidney failure, and bone lesions are some of the symptoms observed in advanced MM stages [10,11,12].

In 2014, the International Myeloma Working Group (IMWG) updated MM diagnostic criteria for MM to allow for earlier diagnosis and the initiation of treatment prior to the occurrence of organ damage. Important progress has been made since, with significant improvements in survival [11,12]. Nevertheless, further efforts and additional resources are needed to successfully treat MM, particularly in the relevant percentages of unfit and frail patients who do not tolerate and/or recover from systemic therapy and require modifications of treatment protocols. Evaluating the functional health status of patients with MM is as important as that of myeloma-related risk factors. Indeed, patients aged ≥ 65 and those with long-term disabilities are known to be highly heterogeneous and require proper assessment strategies to define their frailty profile and most appropriate treatments. Usually, they do not participate in clinical trials and are not eligible to receive treatments of choice, including hematopoietic stem cell transplantation, myeloablative chemotherapies, and enhanced supportive care, but undergo unconventional therapies. To provide strategies of potential use in the selection of relevant therapeutic options, in this review, we analyze treatment protocols recommended for “frail” and older patients with MM.

## 2. Frailty: Definition, Categorization, and Impact on Disease Outcomes

Aging is a gradual and progressive physiological process, associated with several modifications in vital organs and metabolism and increased susceptibility to multiple vulnerabilities. About one-third of newly diagnosed MM patients are >75 years old at diagnosis. Multiple comorbidities, frequently observed in older adults, including chronic kidney disease, osteoporosis, diabetes, obesity, high blood pressure, and various forms of heart disease, may contribute to confounding the clinical picture and delaying diagnosis, which is often reached when MM-related organ dysfunction is already present [13].

While different subgroups of older and frail patients may be identified based on specific characteristics, such as functional status, cognition, socio-economic factors, nutritional status, independence in daily activities, and fitness, they overall represent a poorly investigated population in clinical trials. Indeed, due to the normal decline of organ and system functions, older adults are at a higher risk of experiencing iatrogenic adverse events [14]. Therefore, extensive expertise and comprehensive geriatric assessment (GA) are required to correctly identify personalized management and therapy and to predict tolerance and adverse outcomes [15].

Several GA scoring systems have been proposed. The IMWG has designed a tool to stratify patients considering age, comorbidities, cognitive, and physical conditions [14,16]. The Freiburg Comorbidity Index (Initial Myeloma Comorbidity Index—I-MCI) is based on Karnofsky performance status and on lung and renal impairment [17]. The Revised Myeloma Comorbidity Index (R-MCI) has extended its assessment to disease cytogenetics [18,19]. Moreover, the Geriatric Assessment in Hematology (GAH), a 30-item scale directed at older patients diagnosed with hematological malignancies, dedicates special attention to psychometric evaluation, including, in addition to standard comorbidities, polypharmacy, gait speed, mood, daily activities, health status, nutrition, and mental status [17,20]. Additional GA tools and scoring systems have also been developed more recently [21,22,23].

The goal of personalized treatment in older patients should be to provide a therapy that effectively prolongs survival without affecting patients’ quality of life. A balance between benefits in disease control and the conservation of quality of life, physical independence, and preventing hospitalization should be pursued. A global assessment should also include palliative therapies and hospice care. In this vein, multiple recommendations have been formulated to guide the use of specific drugs [15,24]. As an example, dexamethasone plays a key role in MM treatment. However, despite its effectiveness, high-dose therapy may be associated with neurological and psychiatric disorders, iatrogenic diabetes mellitus, gastrointestinal bleeding, higher risk of infection, and glucocorticoid-induced osteoporosis [15]. Moreover, its cardiovascular impact is non-negligible, particularly in frail and elderly patients. Prolonged use is related to the development or worsening of hypertension, fluid retention, and metabolic dysregulation, increasing the risk of arrhythmias and the decompensation of prior chronic heart failure [25]. The toxicity profile of this key player in MM treatment has led to the development of low-dose dexamethasone administration schedules intended for older and unfit patients. The formulation of fixed-duration treatment schedules to reduce toxicity while preserving effectiveness is currently under investigation [15,26,27].

## 3. Choice of Treatment Regimen in Older MM Patients Ineligible to Receive Transplantation

Frailty status represents a key factor in the management of older patients with MM. Those above 75 years of age are immediately classified as intermediately fit or frail [28] since the use of aggressive treatments in these patients may lead to negative outcomes. Indeed, it has been demonstrated that frail older patients with MM have worse progression-free (PFS) and overall survival (OS) in comparison to fit individuals and show higher rates of infection and treatment toxicity [19,29]. Therefore, personalized treatments should be administered according to the assigned frailty score. However, the use of different tools leads to high variability in the definition of frailty in MM, with prevalence rates ranging between 17.2% and 66.0% [21], and treatment is usually determined on a case-by-case basis. Nevertheless, patients aged over 75 are typically deemed transplant-ineligible.

The recognition of the importance of frailty status has led to reconsiderations of MM treatment. While Melphalan and prednisone with or without bortezomib or thalidomide (MPT, VMP, or MP) previously represented the standard of care for these patients [17], VRd (bortezomib–lenalidomide–dexamethasone) triplet therapy has subsequently been utilized based on the results of the SWOG S0777 phase 3 study, showing that the addition of bortezomib to lenalidomide–dexamethasone results in improved PFS and OS, irrespective of age [30,31]. These results were also confirmed by O’Donnell and colleagues, who reported that in patients older than 75 years requiring a reduction in the intensity of the VRd regimen, modifications to doses and schedules did not compromise treatment effectiveness [32]. Indeed, their data, obtained from a phase 2 study, underline that the modified VRd regimen was not only well tolerated and highly effective but also more appropriate for older patients [33].

More recently, the addition of anti-CD38 monoclonal antibodies (mAb), such as daratumumab, to similar drug combinations has led to the achievement of good and long-lasting responses in several patient groups. Treatment with DRd (daratumumab–lenalidomide–dexamethasone) or D-VMP (daratumumab plus bortezomib, melphalan, and prednisone) [11,13] has resulted in longer than 4/5-year survival [34]. Therefore, DRd and D-VMP should now be considered the gold standard first-line therapy for elderly and/or frail patients. A phase 3 clinical trial supported the first-line use of DRd for patients with MM ineligible for transplantation, regardless of frailty status [35]. Moreover, a further subgroup analysis, including fit, intermediate, and frail patients, has demonstrated that treatment with DRd results in improved PFS, OS, and minimal/measurable residual disease (MRD), as compared to Rd alone, regardless of age and frailty [36,37]. Remarkably, approval of subcutaneous daratumumab—which is considered non-inferior to intravenous formulation in terms of efficacy and pharmacokinetics and provides an improved safety profile also in relapsed/refractory MM patients—has marked a turning point in the management of frail patients with MM [38], with positive effects on patients’ quality of life as well [39]. Other studies have also underlined both a prolonged OS [40] and a 4-fold-higher MRD-negativity rate in patients treated with D-VMP compared to those treated with VMP alone, even in a frail subgroup [41,42,43]. However, DRd appears to be preferable to DVMP, except for patients with severe renal impairment and/or a recurrent history of thrombosis.

Most recently, the EMA has approved the use of isatuximab plus VRd for newly diagnosed MM in transplant-ineligible settings based on the results of a phase 3 trial (IMROZ ClinicalTrials.gov number, NCT03319667). Facon et al. demonstrated that the addition of the anti-CD38 monoclonal antibody isatuximab to the VRd regimen slowed disease progression in patients with MM from 18 to 80 years of age [44].

Comparing isatuximab plus VRd and VRd alone, a better PFS at 60 months was in fact highlighted in an isatuximab-VRd group compared with a VRd group (63.2% vs. 45.2%), with a similar safety profile [44].

Notably, however, the use of multiple-agent treatments in MM patients over 75 has also been shown to be associated with higher toxicity [45]. Within this context, it is important to consider that high-dose dexamethasone, included for almost 40 years in several drug combinations for MM treatment [46], is associated with a variety of adverse effects, including neurological disorders and, most importantly, increased susceptibility to infections, cardiovascular complications, and secondary diabetes mellitus [47,48,49].

Therefore, to allow older, intermediately fit patients to receive and tolerate sufficiently prolonged dexamethasone treatments, dose adjustments based on clinical, functional status and comorbidities have been proposed, aimed at managing potentially occurring adverse events [36]. In this context, Stege and colleagues [50] reported that in frail patients with a median age of 81 years, a low-dose dexamethasone treatment of up to 10 mg every 2/4 weeks had potential advantages and was not less effective. Moreover, in these patients, short-term treatments would be preferable [47,48], as also demonstrated by a randomized phase 3 study where a dose/schedule-adjusted Rd treatment without dexamethasone was used in the maintenance (Rd-R) of intermediately fit patients [26]. Other trials are underway aimed at determining the possibility of omitting dexamethasone in older and frail patients. Preliminary data from the IFM 2017_03 trial anticipate that in frail NDMM patients, the early discontinuation of dexamethasone (after two cycles) in the context of daratumumab and lenalidomide treatment is associated with effective and durable responses with a manageable safety profile [15].

The choice of treatment for relapsing elderly patients is even more challenging. A series of factors, including front-line treatment, patient age, long-term side effects, comorbidities, and the presence of cytogenetic alterations, needs to be considered [51,52]. Data from relapsed/refractory MM frail patients enrolled in the CANDOR trial have shown that daratumumab in combination with Kd56 (D-Kd) is effective and safe and results in improved PFS without additional toxicity [53]. Importantly, cardiac failure has been shown to occur less frequently in frail patients treated with dexamethasone [53].

Table 1 summarizes suggestions for the tailored treatment of patients with MM, with different levels of frailty or defined co-morbidities.

Altogether, the data available from real-world practice regarding older and frail patients are far from optimal. A retrospective analysis of real-world outcomes in about 5000 MM European patients has underlined that only 61% of these patients reach second-line treatment, and only 38% reach the third line [54]. Patients aged ≥ 70 years remain at the highest risk of early mortality in the presence of a variety of aging-related vulnerabilities, resulting in a highly heterogeneous population with diverse clinical outcomes. Therefore, tailored therapeutic approaches are required, carefully balancing effectiveness and treatment burden.

## 4. CAR-T Cell Therapy in Older Patients

Most recently, MM treatment has been revolutionized by the application of targeted therapies, including those with chimeric antigen receptor (CAR) T cells, currently also used in relapsed or refractory MM [55]. Data from the phase 3 CARTITUDE-4 clinical trial have shown that the administration of CAR-T cells targeting the B cell maturation antigen (BCMA) reduces the risk of disease progression or death by 74% compared to two standard-of-care regimens in patients with relapsed and lenalidomide-refractory MM who have previously received one to three lines of treatment [55].

The safety and effectiveness of CAR-T have also been evaluated in elderly patients with R/R MM [56]. Tolerable safety and reasonable effectiveness have been observed [56], although frailer patients have higher treatment-related mortality. Moreover, all grades of immune-effector-cell-associated neurotoxicity syndrome (ICANS) are slightly higher in the elderly population, even if no differences in grade 3 or higher ICANS are observed [56]. Nevertheless, reports on the use of CAR-T cell therapies in older patients are still scarce, the most relevant question being, “How old is too old for CAR-T cell therapies in multiple myeloma?” [57]. Clearly, although age, per se, should not automatically exclude patients from such treatments, related toxicities have to be considered, including not only the most severe events but also low-grade ones impacting quality of life and clinical outcomes.

As underlined by Rajeeve and Usmani [57], to correctly manage treatment-related toxicities, it is critically important to define a correct sequence of therapies. However, to be eligible for CAR-T clinical trials, patients have to meet the strict criteria of performance status and homeostasis regulation that were established to de-risk the development of treatment-related toxicities. Unfortunately, very few elderly patients meet these requirements. Accordingly, the oldest patients included in two important CAR-T phase 2 clinical trials, namely, KarMMa [58] and CARTITUDE-1 [59], were 78 and 70 years old, respectively. Since the median age at MM diagnosis is 69 years, while the median age of patients admitted to phase 2 clinical trials is 61 years, clearly, older patients tend to be excluded from CAR-T therapy. However, interestingly, retrospective meta-analyses underline that anti-BCMA CAR-T cell therapy in older patients with relapsed MM is as safe and effective as in younger patients [60]. Consistently, in a cohort of 83 patients, including 22 ≥70-year-olds, Reyes et al. observed that CRS and ICANS rates, days to absolute neutrophil count recovery, and the incidence of hypogammaglobulinemia were comparable in older and younger patients, without specific geriatric complications [61]. An optimistic view of CAR-T cell therapy in older patients also emerges from a post hoc analysis of the ZUMA-1 trial, documenting similar clinical effectiveness and safety to that of younger patients [62].

On the other hand, Davis et al. reported that although older/frail patients with RRMM treated with CAR-T cell therapy in a real-world setting do not show excess high-grade toxicities, therapy is associated with lower efficacy, as compared to non-frail patients [56].

Taken together, these data support the concept that advanced chronological age, per se, cannot represent an absolute contraindication for CAR-T cell therapy and that physiologic age should instead be assessed as in defined US therapy programs. As an example, Yates et al. used a geriatric-assessment-guided multidisciplinary clinic (GA-MDC) as a tool to evaluate and select older adults for CAR-T cell therapy and to provide an effective risk stratification [63]. Therefore, it would be desirable to enclose geriatric assessments and frailty scores into cellular therapy risk assessment to standardize the selection processes, thereby possibly increasing the number of older patients with MM eligible for CAR-T cell therapy.

## 5. Bispecific Antibodies

Bispecific antibodies (BsAbs) are effectively used in RRMM treatment to promote the elimination of malignant cells using immune effector cells [64]. They include bispecific T cell engagers (BiTEs) and DuoBodies. The main targets are BCMA, G protein-coupled receptor class C group 5 member D (GPRC5D), and Fc receptor-like 5 (FcRH5) (Table 2).

BiTEs combine into one molecule, with a mAb binding a tumor-associated surface antigen and another binding CD3 in the absence of an Fc fragment. They mainly trigger CD8+ T cell cytotoxic activity against targeted tumor cells [65]. However, while they easily penetrate tumors thanks to their small size, they are usually characterized by a short half-life, requiring frequent/continuous infusion [66].

DuoBodies represent evolutions of BiTEs and include two different single-chain variable fragments linked by an Fc domain mediating antibody-dependent cytotoxicity (ADCC), phagocytosis, and complement-dependent cytotoxicity. Their increased half-life also allows intermittent dosing [67].

**Table 2 cancers-17-00944-t002:** BsAbs used in the management of patients with R/R MM.

Targets	BsAbs	Study	Patient Number	Dosing Schedule/Efficacy (Ref.)
BCMA	Teclistamab (JNJ-64007957)	Phase 1/2 MajesTEC-1 trial (NCT04557098); (NCT03145181)	165 Patients ≥ 75 yr: 24 (14.5%).	Subcutaneous injection: 0.06 mg–0.3 mg–1.5 mg/kg once weekly. Deep and durable response [68,69].
	Elranatab (PF-06863135)	Phase 2 MagnetisMM-3 trial (NCT04649359cohort A)	123 Patients ≥ 75 yr: 24 (19.5%).	Subcutaneous injection: 12–32–76 mg once weekly. Good efficacy and safety [70,71].
	Linvoseltamab (REGN5458)	Phase 1/2 LINKER-MM1 trial (NCT03761108)	200 mg.: 117 Patients ≥ 75 yr: 31 (26.4%).	Intravenous injection: 5–25–200 mg once weekly. Consistent efficacy across high-risk subgroups and induced responses in pts who progressed on 50 ng [72,73].
	Alnuctamab (CC-93269)	Phase 1 trial (NCT03486067)	73 Patients ≥ 75 yr: n.a.	Target dose: subcutaneous injection of 30 mg once weekly (cycle 1 to 3) every other week (cycle 4 to 6) and every 4 weeks from cycle 7. Favorable safety profile [74].
	ABBV-383 (TNB-383B)	Phase 1 trial (NCT03933735)	124 Patients ≥ 75 yr: n.a.	Intravenous injection: 60 mg every three weeks. Good tolerance and durable response [75,76].
	REGN-5459	Phase 1/2 trial (NCT04083534)	43 Patients ≥ 75 yr: n.a.	Target dose: intravenous injection of 480 mg once weekly. Acceptable safety/tolerability [77].
GPRC5D	Talquetamab	Phase 1/2 MonumenTAL-1 trial. (NCT03399799); (NCT04634552)	375 Patients ≥ 75 yr: n.a.	Subcutaneous injection: 0.4 mg/kg once weekly or 0.8 mg/kg every other week, with step-up doses. The safety profile was consistent with previous results [78].
	Forimtamig (RG6234)	Phase 1 trial (NCT04557150)	108 Patients ≥ 75 yr: n.a.	Intravenous injection dose range: 6–10,000 µg (51 pts). Subcutaneous injection dose range: 30–7200 µg (57 pts). High response rate across all tested doses for both IV and SC dosing [79].
FcRH5	Cevostamab	Phase 1 trial (NCT03275103)	160 Patients ≥ 75 yr: n.a.	Intravenous infusion in 21-day cycles. In the single-step-up cohorts, the step dose (0.05–3.6 mg) is provided on C1 Day (D) 1, and the target dose (0.15–198 mg) on C1D8. In the double-step-up cohorts, the step doses are provided on C1D1 (0.3–1.2 mg) and C1D8 (3.6 mg) and the target dose (60–160 mg) on C1D15. In both regimens, the target dose is provided on D1 of subsequent cycles. Cevostamab is continued for a total of 17 cycles unless progressive disease or unacceptable toxicity occurs. Clinically meaningful activity and no increase in CRS rate [80].

Abbreviation: n.a.: not available.

These T cell-redirecting therapies are associated with >60% ORR and a median PFS of around 1 year [67]. However, even if first reports are encouraging, the use of BsAbs requires a careful examination of patients > 75 years old, particularly because of potential immune adverse events due to excessive cytokine production. Indeed, Dieterle et al. successfully used teclistamab—a humanized BCMAxCD3 IgG-like BsAb characterized by Fc mutations that stabilize and minimize its immunological effector functions—on octogenarian patients with RRMM, with promising efficacy and reasonable safety [81]. However, the frequent infectious events evidenced in clinical trials negatively impact patient management [82], and preventive measures are required to mitigate this risk, particularly in frail and elderly patients.

## 6. Treatment of Complications and Adverse Events

Frailty often results in a high incidence of adverse events and treatment discontinuation. Therefore, supportive care, prevention, and early identification of complications are crucial for treatment success. Bone disease should be conservatively managed with calcium and vitamin D supplementation; bisphosphonates; and anti-RANKL antibodies, such as denosumab, but may require radiotherapy or orthopedic surgery [13,24]. Infections are a major cause of death among older patients with MM due to multiple factors, including the disease itself, corticosteroid and treatment-related immunosuppression, and neutropenia. Vaccine prophylaxis against influenza, pneumococcal and varicella-zoster virus infections, and pharmacological prophylaxis should be judiciously added to the ongoing treatment. Anemia, present as a prior comorbidity or MM-related, could also affect quality of life and impact old and frail patient equilibrium. This condition may worsen with the initiation of chemotherapy and should be managed by supporting patients with iron supplementation in case of deficiency, erythropoietin-stimulating agents, and red cell transfusions [26].

Clinical signs and symptoms of pain and neuropathy should be carefully evaluated in MM patients. After the initial assessment, pain might represent a warning sign because of its relationship with disease progression or drug-related adverse events [83]. In particular, pain might present in acute and/or chronic form. Its etiology is often multifactorial and associated with bone disease, fractures, nerve compression, peripheral neuropathy, and infectious [VZV] neuralgias. The management of this disabling condition should be patient and symptom-tailored, considering the patient’s characteristics and comorbidities, such as chronic kidney disease. In 2018, guidelines for the management of pain in cancer patients (https://www.who.int/publications/i/item/9789241550390, accessed on 17 June 2024) were provided by the World Health Organization. Depending on its etiology, pain could be treated pharmacologically with corticosteroids; acetaminophen; neuromodulators, including pregabalin and gabapentin; and opioids. Tricyclic antidepressants, such as serotonin uptake inhibitors, and local anesthetics, including lidocaine and capsaicin, could also be of use [13,83,84]. More specifically, bone-active agents, including bisphosphonates and denosumab, or interventional procedures, such as radiotherapy, vertebroplasty, neurolysis, radiofrequency neurotomy, and spinal cord decompression, should be evaluated [85]. Most importantly, frailty and fitness in patients with MM should be considered dynamic features potentially modified by the initiation of treatment and supportive care [86,87]. Therefore, a flexible approach should be adopted to face treatment modification and to warrant patient-tailored management [14]. Notably, ongoing prospective clinical trials are analyzing the clinical significance of the application of a dynamic frailty assessment to guide therapeutic decisions [88] (https://classic.clinicaltrials.gov/ct2/show/NCT06099912, accessed on 17 June 2024).

## 7. Conclusions

Our analysis of published studies suggests that, at least in intermediately fit, elderly, newly diagnosed MM patients, treatment intensity during continuous therapy can be de-escalated without a negative impact on outcomes. However, in subsequent phases, the choice of best therapeutic options, enabling the reduction of treatment-related toxicities and preserving a good quality of life, is more difficult since treatment effectiveness may indeed be affected by the presence of co-morbidities, often related to advanced age, and by defective compliance [57]. As a result, few older and frail patients are currently enrolled in pivotal clinical trials, and few data are presently available on appropriate management (Table 3).

Within this context, it is remarkable that in a large majority of clinical studies, analysis of potential patient frailty is limited to age reports, without mentioning the use of available though imperfect evaluation tools.

On the other hand, while emerging targeted treatments, including CAR-T cells and BsAbs, are powerfully expanding the range of available MM therapeutic options, with increasing effectiveness, their application is also accompanied by the occurrence of specific adverse events [88], with potentially greater impacts on frail and older patients. For the safer use of these treatments, standardized frailty assessments are urgently required based not only on clinical scoring but, possibly, also on the characterization of innovative markers and laboratory tests [88]. In their absence, dynamic assessments and patient-tailored treatments should be recommended.

## Figures and Tables

**Table 1 cancers-17-00944-t001:** Tailoring treatments for MM patients with different IMWG frailty levels and co-morbidities (1–18).

	IMWG-FI Fit	IMWG-FI Intermediate Fit	IMWG-FI Frail	Renal Insufficiency	Peripheral Neuropathy	Cardiovascular Disease
First-Line Therapy	DRd, IsaVRd, or D-VMP	DRd (reduced), D-VMP (reduced), or VRd (reduced)	Rd (reduced) or try Dara-based regimens (reduced)	Prefer Bortezomib-based regimens	Reduce or avoid Bortezomib - Prefer Lena-based regimens	Prefer Dara-based therapy - Avoid Carfilzomib - Reduce Dexa
R/R MM Therapy	KRd, IxaRd (if Lena naïve) Or DaraPd, PVd, SVd, or CAR-T, BsAbs	DaraPd (consider reduction), PVd (consider reduction), or SVd	Low-dose therapies: DRd (reduced if Lena-naïve); DaraPd (reduced) - Supportive care	Prefer Dara-based or regimens (without Lena)	IxaRd - Avoid Carfilzomib	Avoid Carfilzomib - Reduce or discontinue Dexa
Lenalidomide	Standard (25 mg/day)	Dose reduction to 15 mg/day	Dose reduction to 10 mg/day	CrCl 30–50 mL/min: 10 mg/day CrCl < 30 mL/min (no dialysis): 15 mg/2 days CrCl < 30 mL/min (yes dialysis): 5 mg/day post-dialysis	No modification needed	No modification - Thrombo-prophylaxis
Pomalidomide	Standard (4 mg as per therapy schedule)	Dose reduction to 3 mg if toxicity	Dose reduction to 3 mg or 2 mg	No modification	No modification	No modification - Thrombo-prophylaxis
Dexamethasone	40 mg/week	Dose reduction to 20 mg/week	Dose reduction to 10 mg/week or early discontinuation	Dose reduction or discontinuation (if concomitant CHF or DM)	No modification needed	Reduction to 20–10 mg/week or early discontinuation (if CHF, arrhythmia, HTN, or ischemia)
Bortezomib	Twice weekly (1.3 mg/m^2^)	Weekly (1.3 mg/m^2^) or reduce to weekly 1.0 mg/m^2^ if neuropathy	Weekly (1.3 mg/m^2^) or Reduce to weekly 1.0 mg/m^2^ or less if neuropathy	Preferred over Lena - No reduction needed	Reduce to 1.0 mg/m^2^ or less; avoid if severe neuropathy	Avoid in significant cardiopathy and monitor closely if pulmonary hypertension
Daratumumab	Standard	Standard	Standard	No modification	No modification	No modification
Carfilzomib	Standard	Dose reduction to 36 mg/m^2^, weekly administration	Avoid if possible	Avoid if CrCl <30 mL/min	Avoid if severe neuropathy	Avoid in pre-existing CHF and severe HTN

Abbreviations: IMWG-FI: International Myeloma Working Group Frailty Index; DRd: daratumumab + lenalidomide + dexamethasone; IsaVRd: isatuximab + bortezomib + lenalidomide + dexamethasone; D-VMP: daratumumab + bortezomib + melphalan + prednisone; Rd: lenalidomide + dexamethasone; Dara: daratumumab; Lena: lenalidomide; Dexa: dexamethasone; R/R MM: relapsed/refractory multiple myeloma; KRd: carfilzomib + lenalidomide + dexamethasone; IxaRd: ixazomib + lenalidomide + dexamethasone.

**Table 3 cancers-17-00944-t003:** Approved immunotherapy options for older, frail patients ineligible for transplantation at initial diagnosis and at relapse.

Newly Diagnosed Patients	
MAIA study [35]	n = 737 pts (D-Rd, n = 368; Rd, n = 369) 396 non-frail pts (D-Rd, 196; Rd, 200) 341 frail pts (D-Rd, 172; Rd, 169) Clinical benefit irrespective of frailty in newly diagnosed, transplant-ineligible patients
ALCYONE [45]	n = 706 pts (D-VMP, n = 350; VMP, n = 356) 391 non-frail pts (D-VMP, 187; VMP, 204) 315 frail pts (D-VMP, 163; VMP,152) Clinical benefit of D-VMP irrespective of frailty in newly diagnosed transplant-ineligible patients enrolled in ALCYONE, regardless of frailty status.
SWOG S0777 [30,31]	n = 460 pts (VRd, n = 235; Rd, n = 225) 91/235 pts in the VRd arm aged ≥65 years Addition of bortezomib to standard lenalidomide–dexamethasone clinically advantageous irrespective of age in previously untreated patients.
ENDURANCE [89]	n = 1087 pts (VRd, n = 542; KRd, n = 545) VRd lite in older pts [32,33] Addition of Carfilzomib to VRd in newly diagnosed multiple myeloma patients is not more effective and characterized by higher toxicity. A modified VRd treatment effective in >65 years old, newly diagnosed, transplant-ineligible patients
CLARION [90]	n = 955 pts (KMP, n = 478; VMP, n = 477). KMP is not more effective than VMP in newly diagnosed patients with multiple myeloma ineligible for transplant, irrespective of age.
HOVON 143 [50]	n = 65 frail newly diagnosed multiple myeloma pts, treated with Ixa-Dara-Dex. High response rate but toxicity and early mortality.
**Treatment Options at Relapse**	
COLUMBA [91]	n = 522 pts (DARA SC, n = 263; DARA IV, n = 259) Similar effectiveness to daratumumab upon subcutaneous or intravenous administration in relapsed or refractory multiple myeloma.
TOURMALINE [92]	n = 722 (IRd, n = 360; Rd, n = 362) Improved PFS upon treatment with IRd than with Rd in patients with relapsed or refractory multiple myeloma.
ASPIRE-ENDEAVOR-ARROW [93]	ASPIRE n = 792 pts (KRd27, n = 396 vs. Rd n = 396) ENDEAVOR n = 929 pts (Kd56, n = 464 vs. Vd, n = 465) ARROW (once-weekly) n = 478 pts (Kd70, n = 240 vs. Kd27, n = 238) Efficacy and safety were consistent across frailty subgroups with KRd27, Kd56, and weekly Kd70 in relapsed and/or refractory MM.
KarMMa study [94]	CAR-T Therapy (n = 128; n = 45 ≥65 years; n = 20 ≥70 years) Durable responses and manageable safety profile in patients with relapsed/recurrent multiple myeloma aged ≥65 years and ≥70 years.

Abbreviations: pts = patients; D-Rd = daratumumab plus lenalidomide and dexamethasone; Rd = lenalidomide and dexamethasone; D-VMP = daratumumab, plus bortezomib, melphalan, and prednisone; VMP = bortezomib, melphalan, and prednisone; VRd = bortezomib + lenalidomide + dexamethasone; Rd = lenalidomide–dexamethasone; KRd = carfilzomib, lenalidomide, and dexamethasone; KMP = carfilzomib–melphalan–prednisone; Ixa-Dara-Dex = ixazomib–daratumumab–low-dose dexamethasone; DARA = daratumumab; SC = subcutaneous; IV = intravenous; IRd = ixazomib–lenalidomide–dexamethasone; KRd27 = (carfilzomib [27 mg/m^2^]–lenalidomide–dexamethasone; Kd56 = carfilzomib [56 mg/m^2^–dexamethasone; Vd = bortezomib–dexamethasone; Kd70 = carfilzomib [70 mg/m^2^]–dexamethasone.

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
