# Peer review of "Challenges in Multiple Myeloma Therapy in Older and Frail Patients"

_cancers, 2025, doi:10.3390/cancers17060944_

Round 1
Reviewer 1 Report
Comments and Suggestions for Authors
The authors have done extensive review of treatment in frail and elderly myeloma population. There are certain things that can be improved.
- The introduction is too extensive and is focused on many other factors than the topic itself which is treatment of frail elderly myeloma patients, a focus on treatment of myeloma patients in this population and the assessments of friality will make it better.
- Data in tables is very well presented, however, in the text, especially in the anti CD38 sections, there is a lot of jumping from one to the other topic, will recommend doing more focused review.
- There is a systematic review (https://www.sciencedirect.com/science/article/abs/pii/S1879406823002254) on use of CAR-T in myeloma patients in older age groups, which has shown good efficacy and comparable safety in older patients compared to younger patients.
Author Response
Comment 1- The introduction is too extensive and is focused on many other factors than the topic itself which is treatment of frail elderly myeloma patients, a focus on treatment of myeloma patients in this population and the assessments of friality will make it better.
Response 1-We thank this reviewer for this comment. However, we deemed it important to provide the potential readership with some background regarding multiple myeloma (MM) pathogenesis and clinical manifestations. Nevertheless, to comply with requirements from this reviewer the “Introduction” section has been substantially shortened with a particular emphasis on challenges associated with the treatment of frail and older patients (lines 61-72).
Comment 2- Data in tables is very well presented, however, in the text, especially in the anti CD38 sections, there is a lot of jumping from one to the other topic, will recommend doing more focused review.
Response 2-We thank this reviewer for her/his appreciation of the article’s tables. The “anti CD38 sections” have been restructured, as requested (lines 143-166) to provide a more focused discussion on these biologicals, including most recently approved treatments
Comment 3- There is a systematic review (https://www.sciencedirect.com/science/article/abs/pii/S1879406823002254) on use of CAR-T in myeloma patients in older age groups, which has shown good efficacy and comparable safety in older patients compared to younger patients.
Response 3-We thank this reviewer for this suggestion. The mentioned systematic review (1) is now cited in the text.
Reviewer 2 Report
Comments and Suggestions for Authors
This is intended as an overview on the challenges in multiple myeloma therapy in older and frail patients, but in my opinion, this is not fully reflected in the contents of the paper. There is too much detail on the pathophysiology of multiple myeloma in the introduction, which is not relevant to the paper's aim. The section on bispecifics also includes irrelevant details. Both figures are irrelevant and not needed. The content of the paper does not provide, in my opinion, practical information that can be used in practice.
Author Response
Comments- There is too much detail on the pathophysiology of multiple myeloma in the introduction, which is not relevant to the paper's aim. The section on bispecifics also includes irrelevant details. Both figures are irrelevant and not needed. The content of the paper does not provide, in my opinion, practical information that can be used in practice.
Response -Introduction and section on bispecific antibodies have been substantially shortened to eliminate unnecessary details. Figures have also been deleted, as suggested. Most importantly, a new table (Table 1), including recommendations for the treatment of frail patients with MM emerging from specific literature has been included in the revised paper, as requested.
Reviewer 3 Report
Comments and Suggestions for Authors
This is an important manuscript that tackles the problem of myeloma therapy in frail patients. This is a highly relevant question, as most myeloma patients are diagnosed above age of 65, frequently with comorbidities. These patients may not enter clinical trials due to their frailty and comorbidities, therefore many of the published protocols present drug dosing that is not applicable for them. Frailty may also be a prognostic factor more significant than genetic alterations associated with myeloma.
The manuscript attempts to address this problem, however, in its present form, it does not give the much-needed help to interested readers. As myeloma-treating clinicians and investigators, a more detailed presentation of the necessary dose modifications (drug doses, drug scheduling, drug delivery options, etc) would be highly appreciated. The manuscript in its present form does not offer this type of assistance. Possibly one or two diagrams or tables that depict these "cruthes" would certainly make this paper more user-friendly.
A special emphasis should be given to cardiac side effects (not mentioned for high-dose dexamethasone, but highly relevant), to anti-infectious prophylaxis, especially in the face of the present Clostridium pandemic. Moreover, for these patient pool, physical and movement therapy should also be part of their anti-myeloma therapy.
In my opinion, such restructuring of the otherwise well-written manuscript is necessary to achieve maximal effectiveness in influencing myeloma therapy in this ever-growing cohort of frail patients.
Author Response
Comments 1-This is an important manuscript that tackles the problem of myeloma therapy in frail patients. This is a highly relevant question, as most myeloma patients are diagnosed above age of 65, frequently with comorbidities. These patients may not enter clinical trials due to their frailty and comorbidities, therefore many of the published protocols present drug dosing that is not applicable for them. Frailty may also be a prognostic factor more significant than genetic alterations associated with myeloma.
Response 1-We thank this reviewer for her/his appreciation of our work
Comments 2-As myeloma-treating clinicians and investigators, a more detailed presentation of the necessary dose modifications (drug doses, drug scheduling, drug delivery options, etc) would be highly appreciated. The manuscript in its present form does not offer this type of assistance. Possibly one or two diagrams or tables that depict these "cruthes" would certainly make this paper more user-friendly.
Response 2-We thank this reviewer for this suggestion also shared by reviewer no.2 (see above). To comply with these requirements, we have added a new table (Table 1) to the revised version of our manuscript, extensively detailing the personalization of current MM treatments in frail patients or in the presence of defined co-morbidities.
Comments 3-A special emphasis should be given to cardiac side effects (not mentioned for high-dose dexamethasone, but highly relevant), to anti-infectious prophylaxis, especially in the face of the present Clostridium pandemic. Moreover, for these patient pool, physical and movement therapy should also be part of their anti-myeloma therapy.
Response 3-We fully agree that the cardiovascular impact of dexamethasone is non-negligible, particularly in frail and elderly patients. Prolonged use is related to development or worsening of hypertension, fluid retention, and metabolic dysregulation, increasing the risk of arrhythmias and decompensation of a prior chronic heart failure (2) . Dose adjustments may be considered in >= 75 years old patients and according to IMWG-FI (Table 1) (3).Close blood pressure monitoring and periodic cardiac evaluation are essential to balance its benefits against potential cardiovascular harm. The cited side effects are now mentioned as recommended in the text (lines 108-112), and specific references have been added to the bibliography.
Reviewer 4 Report
Comments and Suggestions for Authors
This is a review of treatment available for frail MM patients. However, it seems the review needs to be organized better as some topics are suggested but not elaborated on. The population should be well defined, and the treatment suggested. At the end of the review, new suggestions for personalized treatment and treatment toxicity are discussed. I don't believe that the authors make a valid point for personalized treatment of older patients. Are younger MM patients treated by an individualized approach?
I have several comments:
1. In Fig 1, the authors seem to suggest that the primary genetic aberrations happen in the stem cell stage...I don't believe that is true- the idea so far is that it happens in B cells in the germinal center
2. Generally, MM is considered a hard-to-treat disease, the treatment advances have advanced dramatically in the last 15 years.
3. bone tissue "demolition" is perhaps a bit strong expression
4. The diagnostic difference between MM and SMM was changed in 2014. Please correct accordingly.
5. Since the change in criteria happened over 10 years ago, perhaps it is not necessary to mention the previous criteria.
6. As for comorbidities in older MM patients, one would think that diabetes, obesity, high blood pressure and various forms of heart disease are more common and relevant than vitamin deficiency. Please correct.
7. Do the authors think that CAR T cell therapy would be a good way to treat frail patients? Not older, but frail.
8. The authors should identify a definition of frail patients and stick to it throughout the review.
9. So, based on the presented data, what is the best treatment of frail MM patients and why?
Comments on the Quality of English Language
The paper should be corrected by a native speaker.
Author Response
Comments 1-The population should be well defined, and the treatment suggested. … I don't believe that the authors make a valid point for personalized treatment of older patients.
Response 1-Definition of frailty still represents a critical point in MM treatment. Notably however, although a number of innovative scoring systems have been proposed, as detailed in our study(lines 89-99), the IMWG Frailty Index remains the most widely used (4).In this particular population, initiating a tailored treatment approach, despite the necessity of dose reductions or modification in administration frequency, represents the most effective strategy to synthesize disease control, quality of life preservation and avoiding a discontinuation of the first-line therapy (4). Nevertheless, to comply with criticisms from this and other Reviewers, we have added to our study a new table with recommendations for the treatment of frail patients, as distilled from current literature (Table 1).
Comments 2-In Fig 1, the authors seem to suggest that the primary genetic aberrations happen in the stem cell stage...I don't believe that is true- the idea so far is that it happens in B cells in the germinal center.
Response 2-Figure 1 has been deleted, as required from Reviewer no. 2. Nevertheless, we clearly state in the main text that “Malignant transformation leading to MM clinical onset begins inside lymph nodes’ germinal centers (lines 44-45).
Comments 3- Generally, MM is considered a hard-to-treat disease, the treatment advances have advanced dramatically in the last 15 years.
Response 3-We fully share the opinion of this reviewer. Indeed, on lines 58-61 we state that “In 2014, the International Myeloma Working Group (IMWG) updated MM diagnostic criteria for MM to allow earlier diagnosis and initiation of treatment prior to the occurrence of organ damage. Important progress has been made since, with significant improvements in survival”.
Comments 4-. bone tissue "demolition" is perhaps a bit strong expression
Response 4-We thank this reviewer for her/his comment. The cited wording has been modified into “bone tissue degradation” (line 48), as requested.
Comments 5- The diagnostic difference between MM and SMM was changed in 2014. Please correct accordingly.
- Since the change in criteria happened over 10 years ago, perhaps it is not necessary to mention the previous criteria.
Response 5-We thank this reviewer for these important remarks. Please note, however, that the cited paper (5), also mentioned in our article, did not question the differential diagnosis of MGUS, SMM and MM, but mainly focused on the identification of the large subgroup of SMM patients likely to rapidly develop multiple myeloma, and potentially taking advantage of treatment prior to the presentation of clinical symptoms. In this respect, our study does not contradict those criteria and current understanding (https://www.myeloma.org).
Comments-6. As for comorbidities in older MM patients, one would think that diabetes, obesity, high blood pressure and various forms of heart disease are more common and relevant than vitamin deficiency. Please correct.
Response 6-We thank this reviewer for this comment. Please refer to our reply to Reviewer no. 3. The text has been modified as suggested (lines 78-79 and 108-112).
Comments-7. Do the authors think that CAR T cell therapy would be a good way to treat frail patients? Not older, but frail.
Response 7-At this point in time very few aged and/or frail patients are administered CAR T cell therapies. However, emerging data appear to suggest that they might also take advantage of these treatments. We dedicate a large section of our study to the analysis of these results, but reports are still scarce, and do not allow reliable conclusions as yet (6, 1).A review specifically addressing these issues (1) has now been cited in our study, as also requested by Reviewer no. 1.
Comments-8. The authors should identify a definition of frail patients and stick to it throughout the review.
Response 8-Please refer to our reply to the first comment from this Reviewer.
Comments - 9. So, based on the presented data, what is the best treatment of frail MM patients and why?
Response 9-A new table (Table 1) has been added to the revised manuscript with details on “the best treatment of frail MM patients”, as also required in the initial comment on our study by this Reviewer (see above).
Round 2
Reviewer 4 Report
Comments and Suggestions for Authors
no further comments.